# Tailless/TLX reverts intermediate neural progenitors to stem cells driving tumourigenesis via repression of *asense/ASCL1*

Anna E Hakes, Andrea H Brand*

The Gurdon Institute and Department of Physiology, Development and Neuroscience, University of Cambridge, Cambridge, United Kingdom

**Abstract** Understanding the sequence of events leading to cancer relies in large part upon identifying the tumour cell of origin. Glioblastoma is the most malignant brain cancer but the early stages of disease progression remain elusive. Neural lineages have been implicated as cells of origin, as have glia. Interestingly, high levels of the neural stem cell regulator TLX correlate with poor patient prognosis. Here we show that high levels of the *Drosophila* TLX homologue, Tailless, initiate tumourigenesis by reverting intermediate neural progenitors to a stem cell state. Strikingly, we could block tumour formation completely by re-expressing Asense (homologue of human ASCL1), which we show is a direct target of Tailless. Our results predict that expression of TLX and ASCL1 should be mutually exclusive in glioblastoma, which was verified in single-cell RNA-seq of human glioblastoma samples. Counteracting high TLX is a potential therapeutic strategy for suppressing tumours originating from intermediate progenitor cells.

*For correspondence:
a.brand@gurdon.cam.ac.uk

Competing interests: The authors declare that no competing interests exist.

## Introduction

The underlying mechanisms of glioblastoma initiation and growth have proved challenging to elucidate. This is due, in part, to the extensive molecular heterogeneity of glioblastoma, both between patients and within individual tumours. As such, there are many potential routes to tumourigenesis and the cell fate changes that contribute to glioblastoma initiation and progression remain to be fully elucidated. Cell fates can be altered in many different ways during tumourigenesis, depending upon the combination of genetic mutations present and the tumour cell of origin.

Mouse models have revealed many of the different cell types that can give rise to glioblastoma. In the central nervous system (CNS), tumours have been induced experimentally from differentiated glial cells, glial precursors and neural stem/progenitor cells (*Alcantara Llaguno et al., 2015*; *Alcantara Llaguno et al., 2009*; *Bachoo et al., 2002*; *Chow et al., 2011*; *Friedmann-Morvinski et al., 2012*; *Holland et al., 2000*; *Lindberg et al., 2009*; *Marumoto et al., 2009*). A recent study revealed that astrocyte-like neural stem cells (NSCs) in the SVZ of glioblastoma patients harbour driver mutations that are found in the patient's tumour, suggesting that astrocyte-like NSCs are cells of origin of glioblastoma in humans (*Lee et al., 2018*). Neural lineages become more resistant to glioblastoma transformation as differentiation progresses, supporting stem cells or early progenitor cells as a common source of glioblastoma (*Alcantara Llaguno et al., 2019*). However, it is difficult to state unequivocally which cell type gives rise to tumours in mouse models of glioblastoma due in part to the lack of specific markers and driver lines. For example, both stem and progenitor cells express Nestin (*Chen et al., 2009*) and GFAP labels both stem cells and astrocytes (*Doetsch et al., 1999*). Furthermore, the mechanism through which cells within NSC lineages change identity during tumourigenesis and contribute to tumour aggressiveness remains unclear.

The *Drosophila* CNS has proved extremely valuable for understanding the fundamental principles of cancer (*Deng, 2019*; *Villegas, 2019*). The availability of an unparalleled *Drosophila* genetic toolkit and extensive knowledge of neural cell fate transitions has enabled diverse aspects of tumourigenesis to be investigated. One *Drosophila* model of glioblastoma is based on co-activation of EGFR and PI3K in glial cells (*Chen and Read, 2019*; *Chen et al., 2019*; *Chen et al., 2018*; *Read et al., 2009*; *Read et al., 2013*; *Witte et al., 2009*). This model recapitulates some of the features of glioblastoma, however, co-activation of EGFR and PI3K does not transform NSCs or their progeny. As a result the model does not address the contribution of neural lineages to glioblastoma (*Read et al., 2009*).

High levels of the orphan nuclear receptor TLX (also known as NR2E1, Nuclear Receptor Subfamily 2 Group E Member 1) have been observed in glioblastoma and been shown to correlate with poor patient prognosis (*Park et al., 2010*; *Zou et al., 2012*). TLX is expressed in adult NSCs, where it is required for neurogenesis in both the subventricular zone (SVZ) and the subgranular zone (SGZ) (*Liu et al., 2008*; *Liu et al., 2010*; *Shi et al., 2004*; *Zhang et al., 2008*; *Zou et al., 2012*). TLX is also expressed in glioblastoma stem cells (*Zhu et al., 2014*) and upregulation of TLX promotes gliomagenesis in the mouse SVZ (*Liu et al., 2010*). These results indicate that TLX is an important stem cell regulator both in normal and tumourigenic conditions. However, it is not known how abnormally high TLX levels affect the identity of cells in NSC lineages nor has the cell type vulnerable to TLX overexpression been identified.

In *Drosophila,* different NSC lineages exhibit distinct vulnerabilities to tumour-inducing mutations (*Hakes and Brand, 2019*). The majority of lineages arise from Type I NSCs (*Figure 1A*) that divide asymmetrically to self-renew and generate ganglion mother cells (GMCs), which then undergo terminal division (*Figure 1B*; *Harding and White, 2018*; *Ramon-Cañellas et al., 2019*). A much smaller number of Type II NSCs, by contrast, generate intermediate neural progenitors (INPs) (*Figure 1B'*; *Bello et al., 2008*; *Boone and Doe, 2008*; *Bowman et al., 2008*) that are themselves able to self-renew and produce GMCs. These transit amplifying Type II lineages more closely resemble neural lineages in the vertebrate CNS and provide an opportunity to investigate whether conserved mechanisms regulate how NSCs and their progeny respond to tumourigenic insults.

Here we show that the *Drosophila* TLX homologue, Tailless (Tll), is required to direct the identity of Type II NSCs during development. We found that high levels of Tll are sufficient to initiate tumours from differentiating Type II NSC lineages by directing a cell fate change from INP to NSC. To identify downstream effectors of Tll action, we mapped the genome-wide targets of Tll and identified the proneural gene *asense* as a direct target of Tll repression, both during development and in tumourigenesis. Strikingly, we were able to rescue Tll tumours completely, and restore normal neurogenesis, by re-expressing *asense*. Our results demonstrate a reciprocal relationship between Tll and Asense expression and we hypothesized that this relationship might hold true in glioblastoma. We found that expression of TLX and ASCL1 (human counterparts of Tll and Asense) are also mutually exclusive in glioblastoma, suggesting a potentially conserved route to tumourigenesis.

## Results

### Tailless is necessary for Type II NSC identity and lineage progression

To understand the role Tailless (Tll) plays in the development of Type II NSC lineages, we first assessed its expression pattern. We found that Tll was expressed in Type II NSCs throughout larval development (*Figure 1—figure supplement 1A–C*). We detected *tll* mRNA in Type II NSCs but not in their progeny (INPs) (*Figure 1C*), while Tll protein was present in NSCs and at low levels in newly-born INPs (*Figure 1D–D'*). Tll shares a high degree of homology with human TLX (*Figure 1E*; *Jackson et al., 1998*): their DNA binding domains are 81% identical (94% similarity) and their ligand binding domains are 40% identical (77% similarity). In addition, TLX and Tll bind to the same consensus DNA sequence (*Yu et al., 1994*) and recruit conserved cofactors, such as Atrophin, via their ligand binding domains (*Wang et al., 2006*; *Zhi et al., 2015*). TLX is expressed in the neurogenic regions of the adult mouse brain (*Liu et al., 2008*; *Niu et al., 2011*; *Shi et al., 2004*; *Zhang et al., 2008*). In the SVZ, TLX is detected in NSCs and their progeny, intermediate progenitor cells (*Figure 1F*; *Li et al., 2012*; *Obernier et al., 2011*), which is very similar to the expression pattern of Tll in *Drosophila* Type II lineages (*Figure 1G*).

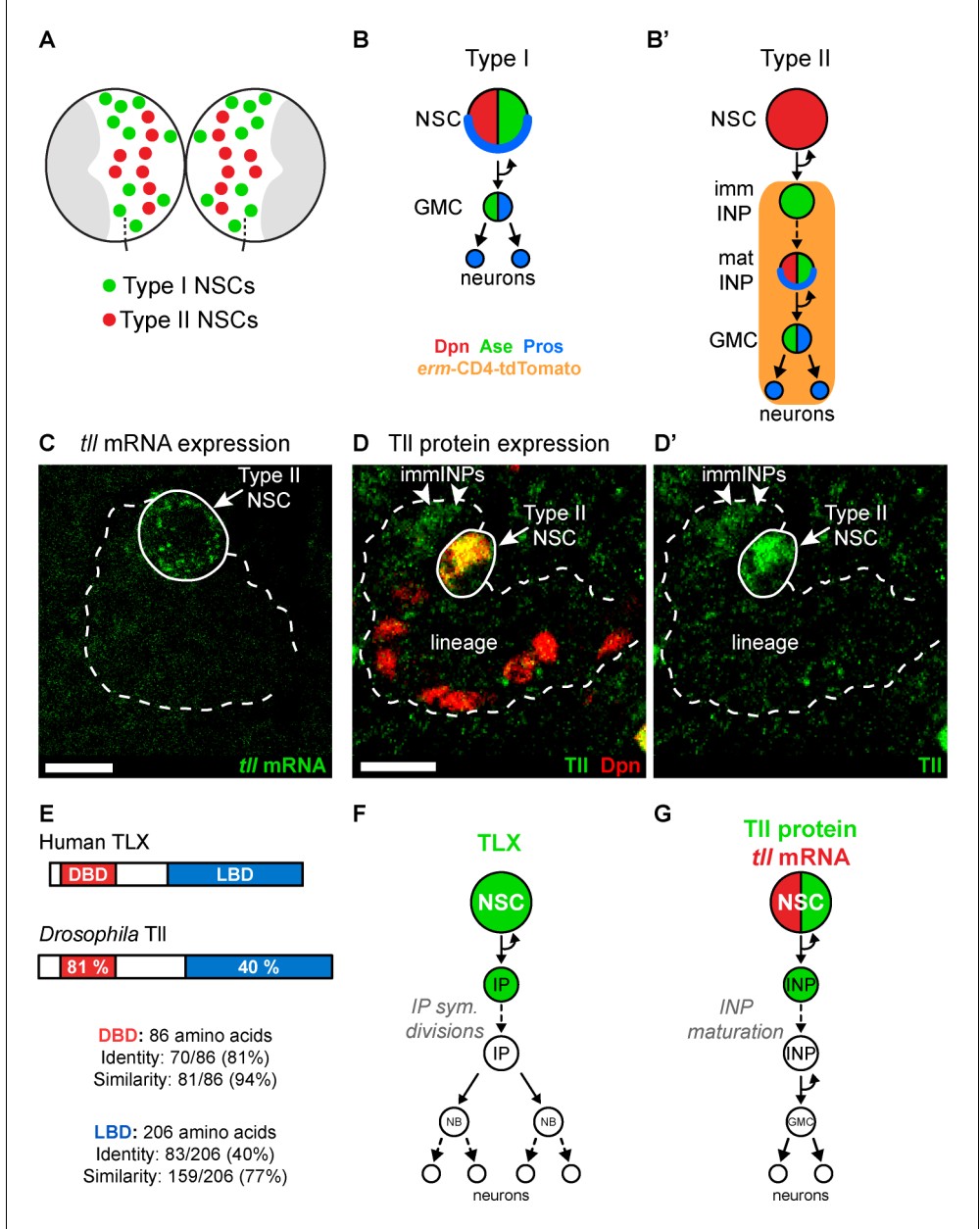

**Figure 1.** Tll is expressed in *Drosophila* Type II NSCs. (**A**) Schematic showing the position of the eight Type II NSCs (red) in each brain lobe. The majority of stem cells in the *Drosophila* brain are Type I NSCs (green). The optic lobes, which generate the adult visual processing centre, are shown in grey. (**B–B'**) Schematics showing the expression of cell fate markers in (**B**) Type I and (**B'**) Type II lineages. NSC: neural stem cell; imm INP: immature intermediate neural progenitor; mat INP: mature intermediate neural progenitor; GMC: ganglion mother cell. (**C**) RNA FISH shows *tll* mRNA (green) expression in Type II NSCs (solid outline) but not in their lineages (dotted outline). Type II lineages were identified by *pntP1*-GAL4 > *mCD8-GFP* expression in the central brain at wandering third instar larval stage. (**D–D'**) Immunostaining for Tll (green) shows strong expression in Type II NSCs (Dpn+ (red), solid outline) and weak expression in Dpn- immature INPs (immINPs, arrow heads). Mature INPs (small Dpn+ cells in the lineage) do not express Tll. Type II lineages were identified by *pntP1*-GAL4 > *mCD8-GFP* expression in the central brain at wandering third instar larval stage. (**E**) Amino acid conservation between human TLX and *Drosophila* Tll. (**F**) Schematic showing that TLX (green) is expressed in NSCs and intermediate progenitors (IPs) in SVZ of the adult mouse brain (***Li et al., 2012***; ***Obernier et al., 2011***). (**G**) Schematic showing *tll* mRNA (red) and Tll protein (green) expression in *Drosophila* Type II NSC lineages. Single section confocal images. Scale bars represent 10 µm.

The online version of this article includes the following figure supplement(s) for figure 1:

*Figure 1 continued on next page*

*Figure 1 continued*

**Figure supplement 1.** Tll is expressed in *Drosophila* Type II NSCs.

The enrichment of Tll expression in Type II NSCs suggested a role for Tll in regulating Type II NSC identity or proliferation. We knocked down Tll in larval NSCs using *wor*-GAL4 using two independent RNAi constructs that target different regions of the *tll* coding sequence (*Figure 2—figure supplement 1A*). We scored expression of Deadpan (Dpn), a Hes family bHLH-O transcription factor that is expressed in all NSCs (*Bier et al., 1992*), and Asense (Ase), a proneural bHLH factor expressed in Type I but not Type II NSCs (*Bowman et al., 2008*). Expressing either *tll* RNAi construct resulted in the absence of all Type II NSC in all brains assessed (*i.e.* all NSCs expressed Dpn and Ase) (*Figure 2—figure supplement 1B–C'*). We also generated *tll* null MARCM (Mosaic Analysis with a Repressible Cell Marker) clones (*Lee and Luo, 1999*). We found that Type II lineages were often labelled in wild type clones (*Figure 2—figure supplement 1D–D'*), demonstrating that MARCM clones could encompass Type II NSCs. However, we were unable to recover *tll* null Type II NSC clones, despite mutant clones being visible in other NSC lineages (*Figure 2—figure supplement 1E*). This suggested that *tll* null Type II NSCs underwent a cell fate transition that resulted in the loss of Type II markers. In support of this, quantification of the number of Type II lineages in brains with *tll* null clones revealed a reduction in the number of Type II lineages (*Figure 2—figure supplement 1F*). Furthermore, the number of absent Type II lineages in brains with *tll* null clones was comparable to the number of Type II lineages encompassed in MARCM clones in control brains (*Figure 2—figure supplement 1F*).

Next, we knocked down Tll expression specifically in Type II lineages by driving *tll* RNAi (*tll*-miRNA[s], which effectively knocked down Tll protein (*Figure 2—figure supplement 1G*)) with *pntP1*-GAL4 (*Zhu et al., 2011*) in combination with a 'FLP-out' GAL4 cassette to immortalise GAL4 expression (*Figure 2—figure supplement 2A*) and followed alterations in cell fate. While Dpn expression was unaffected, *tll* knockdown resulted in derepression of Ase in all Type II NSCs (*Figure 2—figure supplement 2B*), suggesting a switch in mode of neurogenesis from Type II to Type I.

To test whether Type II NSCs were transformed into Type I NSCs, we assessed lineage composition and gene expression. Type I NSCs express Dpn and Ase and segregate cortically localised Prospero (Pros) to their daughter cells (GMCs). In GMCs, Ase is expressed and Pros, a pro-differentiation transcription factor, translocates to the nucleus. In contrast, Type II NSCs express Dpn but not Ase or Pros. Type II NSCs give rise to INPs, which express Dpn, Ase and cortical Pros. INPs then generate GMCs that are Ase$^+$ Pros$^+$. In addition, the Ets transcription factor PointedP1 (PntP1) is expressed in Type II NSCs and immature INPs but not in Type I lineages (*Zhu et al., 2011*). Type II lineages can also be labelled by expression driven by a regulatory fragment of the FezF transcription factor *earmuff* (*erm*), which is expressed from INPs onwards (*Pfeiffer et al., 2008*; *Weng et al., 2010*) but not in Type I lineages.

Strikingly, upon *tll* knockdown, no INPs (small Dpn$^+$ Ase$^+$) could be found in Type II lineages and instead GMCs (Ase$^+$ Pros$^+$) were positioned adjacent to the NSCs (compare *Figure 2A–A''* to *Figure 2B-B''*). In 6 out of 10 *tll* knockdown brains, at least one Type II NSC expressed Pros that localised in a crescent at the cell cortex, indicating asymmetric segregation of Pros to daughter cells (*Figure 2B''*). Asymmetric segregation of Pros is a feature characteristic of Type I NSCs and INPs that is never observed in Type II NSCs (*Bayraktar et al., 2010*; *Bello et al., 2008*). Furthermore, expression of Pnt-GFP (*Boisclair Lachance et al., 2014*) in Type II NSCs (*Figure 2—figure supplement 2C–C'*) and the Type II lineage marker, *erm*-CD4-tdTomato (*Han et al., 2011*), was lost in the absence of *tll* (*Figure 2—figure supplement 2D*). We conclude that, in the absence of *tll*, Type II NSC lineages are transformed into Type I lineages which exhibit lower neurogenic capacity due to the lack of INPs (*Figure 2C–C'* and *Figure 2—figure supplement 2E–E'*).

## Tailless tumours can arise from Type II INPs and Type I NSCs

We have shown that Type II NSCs are lost when Tll is downregulated. As a corollary, we hypothesised that ectopic expression of Tll might result in excess Type II NSCs. To test this, we drove Tll expression in INPs and their progeny with *erm*-GAL4 (*Pfeiffer et al., 2008*; *Weng et al., 2010*) in combination with a 'FLP-out' GAL4 cassette to immortalise GAL4 expression (*Ito et al., 1997*;

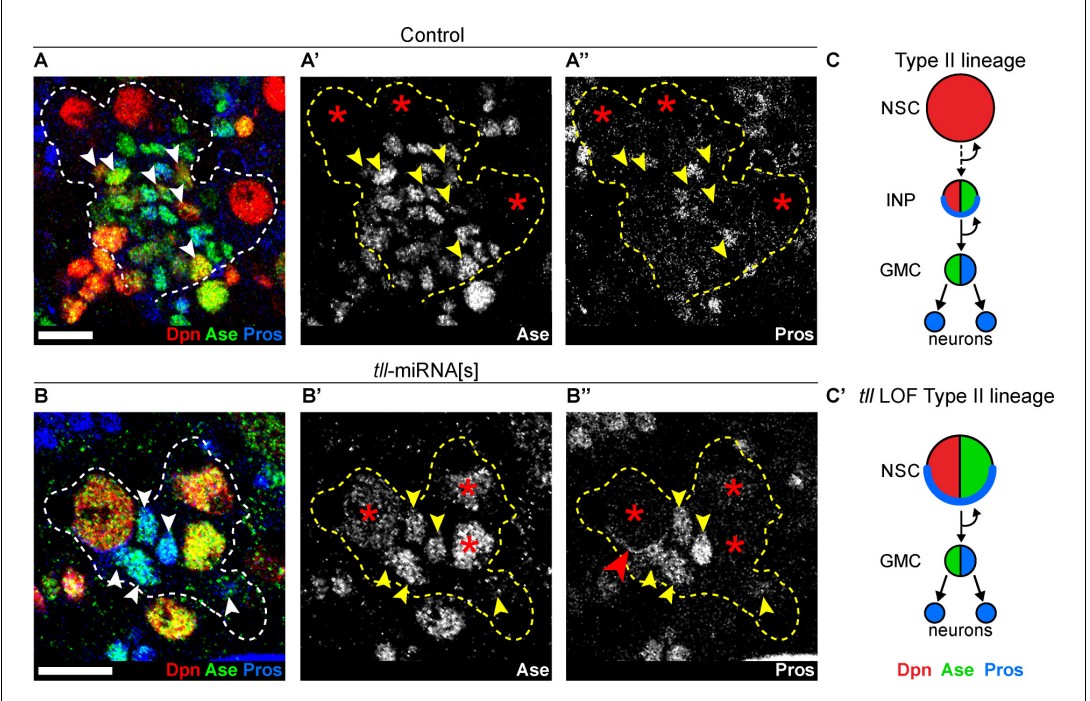

**Figure 2.** Tll is required for Type II NSC fate and lineage progression. (**A–A''**) Control Type II NSCs (Dpn[+] (red) and Ase[-] (green)) generate INPs (arrowheads, Dpn[+], Ase[+] and Pros[-] (blue)). Dotted lines outline three Type II lineages. Red asterisks (*) indicate Type II NSCs. n = 10 brains, dissected at the end of second larval instar stage. (**B–B''**) Upon *tll* knockdown using *pntP1 >act*-GAL4 to drive UAS-*tll*-miRNA[s], Type II NSCs express Ase and generate GMCs directly (arrowheads, Ase[+] and Pros[+]) and exhibit Pros crescents (red arrowhead). Dotted lines outline three Type II lineages identified by *pntP1 >act*-GAL4 driving UAS-*GFP*. Red asterisks (*) indicate Type II NSCs. n = 10 brains, dissected at the end of second larval instar stage. (**C–C'**) Schematic summarising the *tll* loss of function (LOF) phenotype in Type II NSCs. Single section confocal images. Scale bars represent 10 μm.

The online version of this article includes the following figure supplement(s) for figure 2:

**Figure supplement 1.** Tll loss of function in Type II NSCs.

**Figure supplement 2.** Generating an immortalised Type II NSC driver to assess cell fate changes.

*Figure 3A* and *Figure 3—figure supplement 1A–B*). In control brain lobes there are only eight Type II NSCs. Tll misexpression in INPs resulted in a dramatic increase in the number of Type II NSCs, from 8 to 109 ± 12.12 per brain lobe (*Figure 3B–B'* and quantified in *Figure 3C*). We also observed a strong reduction in the number of differentiating progeny in Type II lineages (*Figure 3—figure supplement 1C–D'*). We tested whether Tll expression in more differentiated neural precursors (GMCs) or post-mitotic neurons would also generate excess NSCs but, interestingly, we found that both cell types were resistant to ectopic Tll expression: we observed neither additional NSCs nor neuronal dedifferentiation (*Figure 3D–E'*). Therefore, ectopic Tll expression in Type II lineages promotes NSC fate at the expense of self-renewing cells, specifically INPs (*Figure 3—figure supplement 1E–E'*).

INPs closely resemble Type I NSCs, in that they divide in the same manner and express common cell fate markers (Dpn, Ase and cortical Pros; *Figure 1B–B'*). We tested if Type I NSCs, which are found throughout the CNS, are also vulnerable to high levels of Tll expression. When we expressed Tll throughout the CNS, we observed large tumour-like growths in the adult brain that consisted almost entirely of NSCs (Dpn[+] cells) (compare *Figure 4A–A'*). At larval stages, we found that high levels of Tll resulted in Dpn[+] NSCs in both the central brain and ventral nerve cord, indicating that ectopic Tll can induce tumours from Type I NSCs (*Figure 4B–B'*).

Expression of Tll in Type I NSCs (normally Tll[-], Ase[+]) might convert them to a Type II fate (Tll[+], Ase[-]). To determine if tumour initiation occurred via conversion of Type I to Type II NSCs, we assessed the expression of Ase in tumours in the ventral nerve cord, which normally contains only Type I NSCs. Remarkably, Tll-induced tumours consisted almost entirely of NSCs that were negative for Ase, indicating a Type II-like NSC fate (*Figure 4C–C'*). Consistent with a change in identity to

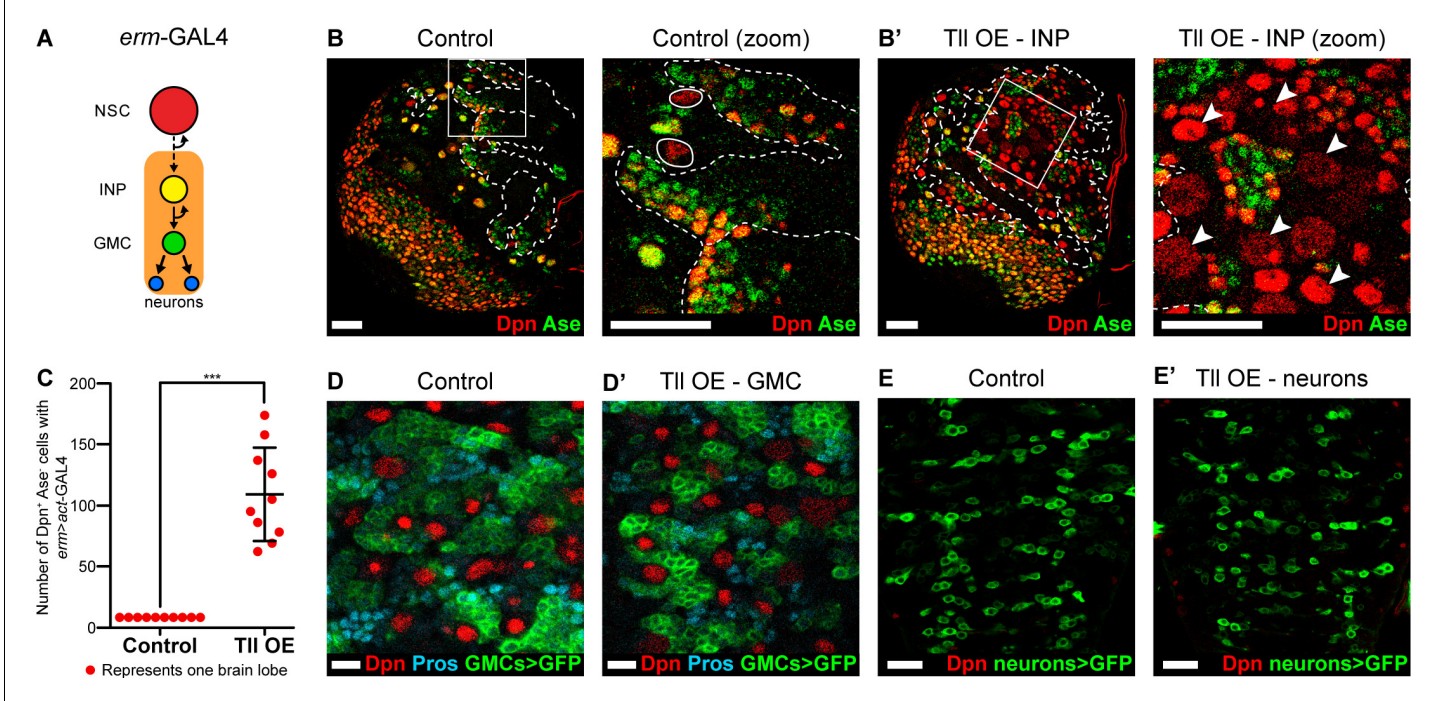

**Figure 3.** Tll overexpression in INPs generates ectopic NSCs. (**A**) Schematic showing the expression of *erm*-GAL4, which begins to be expressed in Type II lineages during the final stages of INP maturation. (**B–B'**) In Control (zoom), solid white outlines indicate Type II NSCs and *erm >act*-GAL4 is expressed in their lineages (dotted white lines). Tll OE in INPs with *erm >act*-GAL4 resulted in a large expansion of Type II NSCs (Dpn$^+$ (red) and Ase$^-$ (green)) in Type II lineages. Arrowheads in Tll OE – INP (zoom) highlight ectopic Type II NSCs. Zoom panels are magnifications of boxed regions in Control and Tll overexpression (OE) – INP. n = 10 brain lobes for Control and Tll. UAS-*tll* expression was restricted to larval stages with *tub*-GAL80$^{ts}$ and brains were dissected at wandering third instar larval stage. (**C**) Quantification of the total number of Type II NSCs (Dpn$^+$ Ase$^-$) in Control or Tll OE *erm >act*-GAL4 brains. Kolmogorov-Smirnov test ***, p<0.001 (p=0.000091). (**D–D'**) Expressing Tll in GMCs (using GMR71C09-GAL4 > *mCD8-GFP* (green)) does not result in ectopic NSCs (*i.e.* no Dpn$^+$ GFP$^+$ cells) nor defects in differentiation, as assessed by Pros (blue) staining. n = 10 brains for Control, n = 12 brains for Tll OE. Brains were dissected at wandering third instar larval stage. (**E–E'**) Expressing Tll in neurons using OK371-GAL4 > *mCD8-GFP* (green) does not result in ectopic NSCs (*i.e.* no Dpn$^+$ GFP$^+$ cells). n = 4 brains for Control and Tll. Brains were dissected at wandering third instar larval stage. Single section confocal images. Scale bars represent 30 μm in (**B, B', E, E'**) and 10 μm in (**D, D'**). The online version of this article includes the following figure supplement(s) for figure 3:

**Figure supplement 1.** Tll overexpression in INPs results in ectopic Type II NSCs.

Type II NSC, the tumour NSCs lacked Pros (*Figure 4D–D'*; *Bayraktar et al., 2010*). Furthermore, a subset of these transformed Type I NSCs generated INPs (*Figure 4—figure supplement 1A–A'*), indicating that Tll is sufficient to induce a switch in NSC identity. Interestingly, the absence of Pros from Tll-induced hyperplasia had been reported previously (*Kurusu et al., 2009*) but had not been linked to a transformation from Type I to Type II NSC fate, or the ectopic appearance of INPs in the ventral nerve cord.

The appearance of the ectopic Type II-like NSCs (Dpn$^+$ Ase$^-$) was associated with a reduction in GMCs and neurons, as assessed by expression of Pros (*Figure 4D–D'*). We conclude that expressing Tll in Type I lineages not only directs a change in NSC identity but also blocks differentiation in these newly transformed lineages, resulting in large NSC tumours comprised of Type II NSCs (*Figure 4E–E'*).

## TLX/Tailless tumour initiation occurs via the reversion of INPs to NSC fate

To determine the cell of origin of Tll-induced tumours, we used G-TRACE (GAL4 technique for real-time and clonal expression) (*Evans et al., 2009*) to follow cell fate transformations within the Type II lineage. G-TRACE reports both current and historic GAL4 expression and so can be used to follow cell lineages (*Figure 5—figure supplement 1A*). *erm*-GAL4 driving G-TRACE labels INPs, but not

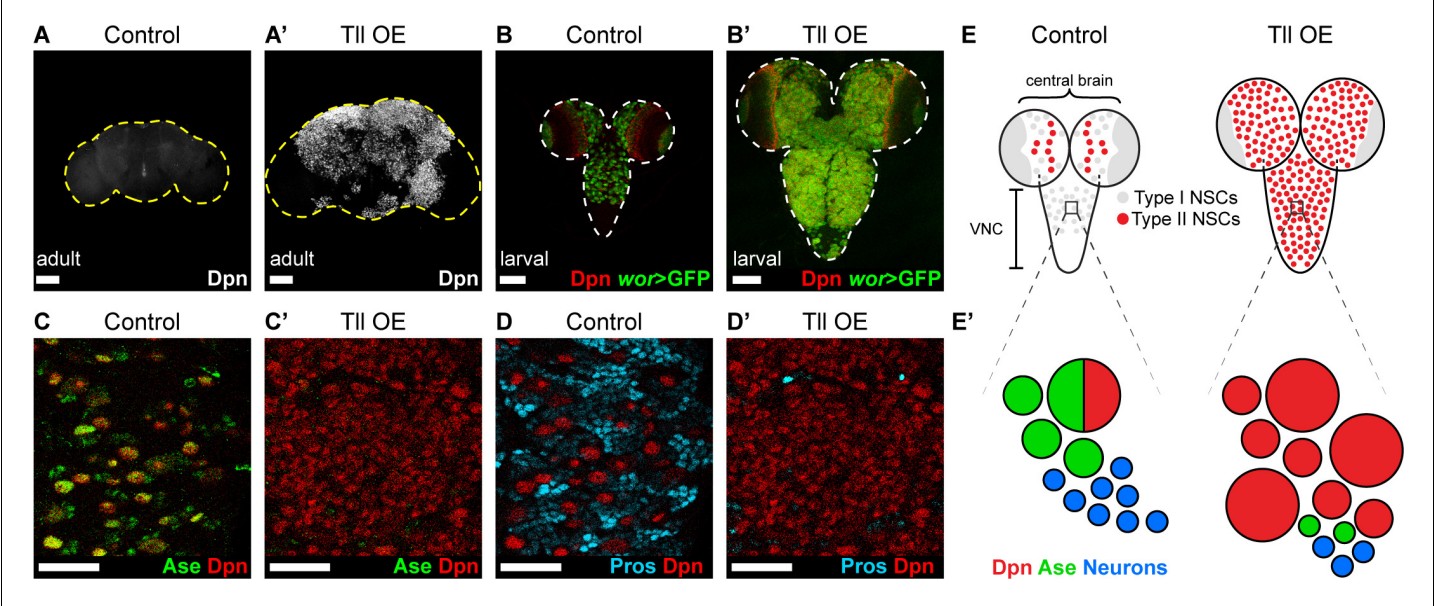

**Figure 4.** Tll can initiate Type II NSC tumours from Type I NSCs. (**A–A'**) Overexpression of *Drosophila* Tll in neural lineages using *wor*-GAL4 resulted in NSC tumours (Dpn[+] (white)) in all adult brains assessed. Control adult brains did not contain any NSCs. *n* = 7 brains for Control and Tll OE. UAS-*tll* expression was restricted to late larval stages with *tub*-GAL80[ts] and brains were dissected from newly-eclosed adult flies. Images are projections over 15 µm (Control) or 17 µm (Tll)). (**B–B'**) Overexpression of Tll during larval development with *wor*-GAL4 resulted in large tumours consisting of ectopic NSCs (Dpn[+] (red) and *wor*-GAL4 >*mCD8-GFP* (green)) in the central brain and VNC of all brains assessed. UAS-*tll* expression was restricted to larval stages with *tub*-GAL80[ts] and brains were dissected at wandering third instar larval stage. (**C–C'**) NSCs in the VNC are Type I (Dpn[+] (red) and Ase[+] (green)) in Control brains. Tll-induced tumours (ectopic Dpn[+] cells) derived from Type I NSCs in the VNC are negative for Ase. UAS-*tll* expression was restricted to larval stages with *tub*-GAL80[ts] and brains were dissected at wandering third instar larval stage. (**D–D'**) Tll tumours in the VNC occur at the expense of differentiating progeny (Pros (blue)). UAS-*tll* expression was restricted to larval stages with *tub*-GAL80[ts] and brains were dissected at wandering third instar larval stage. (**E**) Schematic showing the organisation of Type I NSCs (Ase[+] (grey)) and Type II NSCs (Ase[-] (red)) in Control brains and Tll OE brains. Note that in Control brains the VNC contains only Type I NSCs, whereas Tll OE VNCs contain many ectopic Type II NSCs. (**E'**) Schematic showing transformation of Type I NSC lineages in the VNC to ectopic Type II NSCs when Tll is expressed at high levels. Single section confocal images unless stated otherwise. Scale bars represent 100 µm in (A-B') and 30 µm in (C-D'). *n* = 10 brains for all conditions unless stated otherwise.

The online version of this article includes the following figure supplement(s) for figure 4:

**Figure supplement 1.** Tll is sufficient to induce the generation of INPs from a subset of Type I NSCs.

Type II NSCs, in control brains (*Figure 5A*). Expressing high levels of Tll in INPs resulted in supernumerary Type II NSCs, which had previously expressed *erm*-GAL4 but lacked current expression (*Figure 5A'* and quantified in *Figure 5B*). This would be expected if the cells had originally been INPs (*erm*-GAL4 expressing) and were then transformed into Type II NSCs (*erm*-GAL4 negative). We conclude that Tll expression is sufficient to induce a cell fate change from INP to NSC and our results implicate INP reversion to NSCs as the mechanism of tumour initiation.

To investigate whether human TLX could also initiate tumours from INPs via similar regulatory pathways, we expressed TLX in combination with G-TRACE in INPs. As for Tll, we observed ectopic Type II NSCs with historic *erm*-GAL4 expression only (*Figure 5A''*). Our results indicate that both Tll and TLX can initiate tumourigenesis from neural lineages by reverting INPs to NSCs (*Figure 5C*). Interestingly, neither TLX nor Tll could revert post-mitotic neurons to NSCs, demonstrating that neurons are resistant to tumour initiation (see *Figure 3D* and *Figure 5—figure supplement 2*). We conclude that INPs represent a tumour-susceptible cell type and are the tumour cells of origin for TLX- and Tll-induced tumours in *Drosophila* (*Figure 5D*).

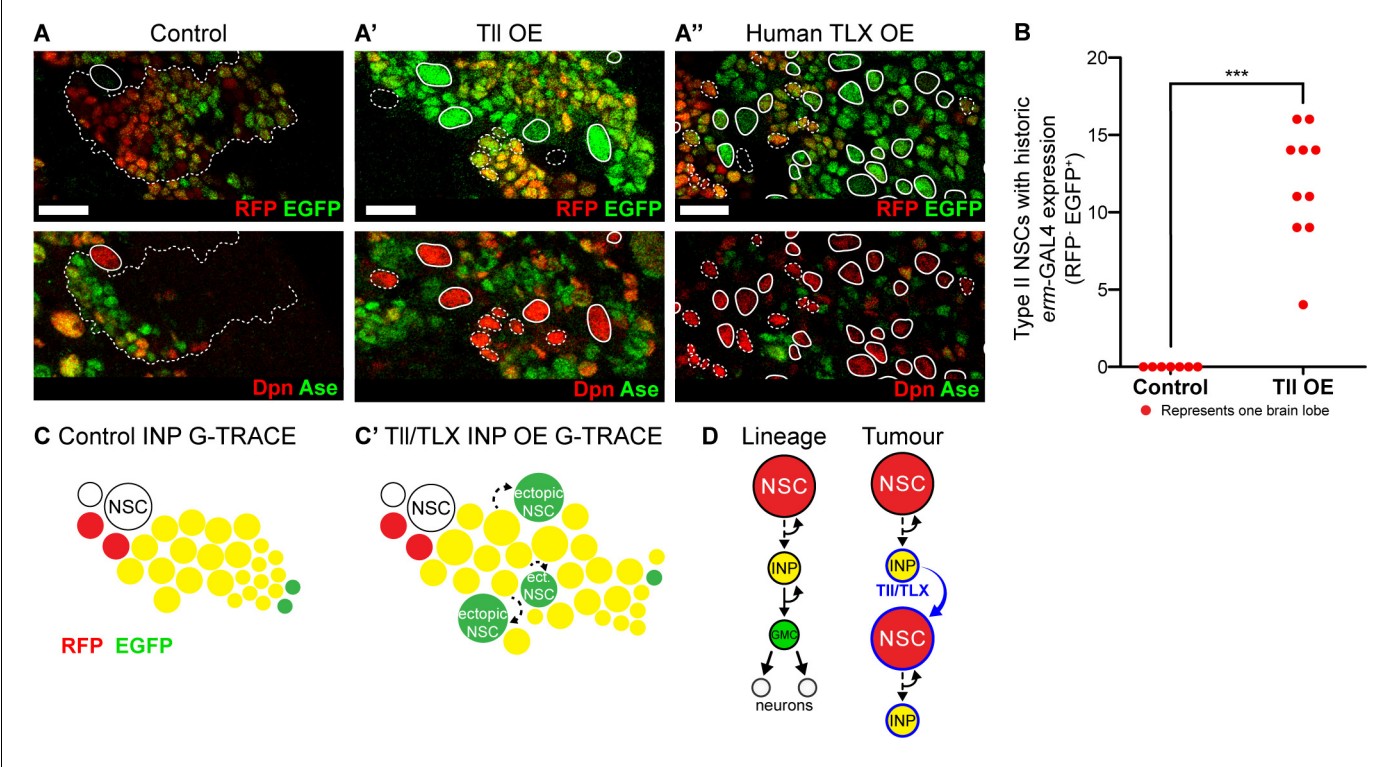

**Figure 5.** Tll/TLX overexpression results in reversion of INPs to NSC fate. (A–A'') G-TRACE reveals current (RFP (red)) and historic (EGFP (green)) *erm*-GAL4 expression (top panels). Dpn (red) and Ase (green) were used to assess the reversion of INPs to Type II NSCs (bottom panels). (A) In Control Type II lineages, NSCs (Dpn$^+$ Ase$^-$, solid outline) are negative for both components of G-TRACE, whereas lineages show transition from RFP to EGFP (dotted outline). Overexpression (OE) of (A') Tll or (A'') human TLX in INPs resulted in ectopic Type II NSCs (Dpn$^+$ Ase$^-$, white outlines) that express the EGFP component of the G-TRACE only (solid outline). Dpn$^+$ Ase$^-$ NSCs with dotted white outline either express neither G-TRACE component (as in Control) or express both RFP and GFP (indicating current expression of *erm*-GAL4). n = 8 brain lobes for Control and n = 10 for Tll and human TLX. Brains were dissected at wandering third instar stage. (B) Quantification of Type II NSCs expressing G-TRACE memory only (*i.e.* Dpn$^+$ Ase$^-$ and RFP$^-$ EGFP$^+$). Kolmogorov-Smirnov test ***, p<0.001 (p=0.000103). n = 7 brain lobes for Control; n = 10 brain lobes for Tll overexpression (OE). Brains were dissected at wandering third instar larval stage. (C–C') Schematic showing the expression of G-TRACE with the INP-specific *erm*-GAL4 in Control brains or with Tll/TLX OE. (D) A model for how Tll/TLX generates ectopic NSCs and, consequently, tumours from INPs. Single section confocal images. Scale bars represent 15 μm.

The online version of this article includes the following figure supplement(s) for figure 5:

**Figure supplement 1.** Tll/TLX overexpression induces reversion of INPs to NSC fate.

**Figure supplement 2.** Expressing TLX in post-mitotic neurons does not result in ectopic NSCs.

## Ase restores progenitor identity and enforces differentiation to block Tailless tumours

We have shown that Tll is both necessary and sufficient to repress Ase expression during development and in tumourigenesis. To investigate if Tll represses *ase* directly, we identified the genome-wide Tll binding sites in vivo using Targeted DamID (TaDa) (*Southall et al., 2013*). We profiled Tll binding in Type II NSCs (16 per brain; approximately 700 NSCs per replicate) (*Figure 6—figure supplement 1A–B*) and identified Tll-binding peaks at 2495 protein-coding genes (see 'Tll TaDa binding targets.xlsx' for full list of Tll targets), including *ase* (*Figure 6—figure supplement 1C*). We conclude that Tll binds *ase* directly and represses its expression to promote Type II NSC fate. However, the loss of Ase alone is not sufficient to induce Type II fate in Type I NSCs (*Bowman et al., 2008*), indicating that Tll acts on additional target genes to mediate this cell fate change and tumour initiation. One potential candidate is Pros, which we found was also bound by Tll in Type II NSCs (*Figure 6—figure supplement 1D*). Pros is known to negatively regulate NSC proliferation (*Cabernard and*

*Doe, 2009*) and is also repressed when Tll is expressed at high levels (*Figure 3—figure supplement 1C–C'* and *Figure 4D–D'*; *Kurusu et al., 2009*).

As ectopic expression of Tll results in repression of *ase* and tumour formation, we investigated whether reinstating *ase* expression might be sufficient to block Tll-induced tumourigenesis. We expressed Ase together with Tll and found, remarkably, that tumourigenesis was completely abolished in all brains analysed (*n* = 10) (*Figure 6A–A''*). Strikingly, not only did Ase prevent the production of ectopic NSCs, it also re-established normal neurogenesis in Type I NSC lineages: the

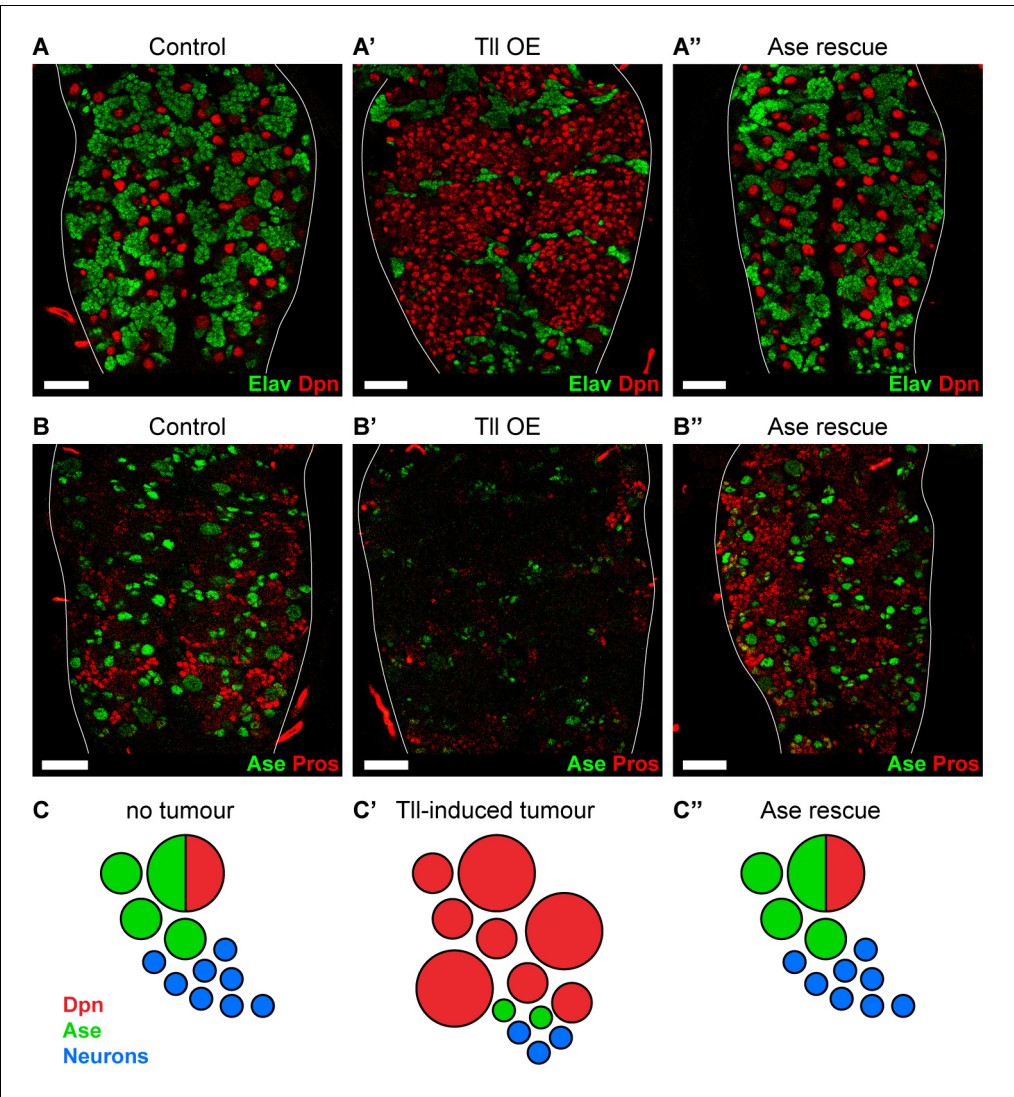

**Figure 6.** Reinstating progenitor identity prevents the formation of Tll tumours. (**A–A''**) Expressing Ase in combination with Tll during larval development using *wor*-GAL4 prevents tumour formation (ectopic Dpn⁺ cells (red)) and restores neuronal differentiation (Elav (green)) in all brains assessed. *n* = 10 brains for all conditions. Brains were dissected at wandering third instar stage. (**B–B''**) Ase (green) rescues Tll tumours by promoting differentiation (Pros (red)). *n* = 9 brains for Control; *n* = 10 brains for Tll overexpression (Tll OE) and Ase rescue. Brains were dissected at wandering third instar larval stage. (**C–C''**) Schematic depicting Type NSC I lineages (**C**) during development, (**C'**) with Tll-induced tumours and (**C''**) with Ase expression in Tll tumours. Single section confocal images. Scale bars represent 30 μm.

The online version of this article includes the following source data and figure supplement(s) for figure 6:

**Figure supplement 1.** Determining Tll target genes in Type II NSCs using targeted DamID (TaDa).

**Figure supplement 2.** Ectopic Ase expression does not repress Tll.

**Figure supplement 2—source data 1.** Tll TaDa binding targets.

production of GMCs and neurons was restored, as revealed by expression of Pros and Elav (*Figure 6A–A'' and B–B''*). However, re-introducing Ase into Tll tumours did not repress the expression of Tll (*Figure 6—figure supplement 2A–A''*) nor does the ectopic expression of Ase turn off Tll in Type II NSCs (*Figure 6—figure supplement 2B–B'*). Therefore, by expressing Ase we were able to reinstate the normal neurogenic programme and prevent tumour initiation by Tll (*Figure 6C–C''*).

## TLX and ASCL1 appear to be mutually exclusive in human glioblastoma

We showed that, in *Drosophila*, Tll represses Ase both during development and in tumourigenesis. In other words, high levels of Tll correspond to low levels of Ase.

In human glioblastoma, high TLX expression is correlated with poor patient prognosis (*Park et al., 2010*; *Zou et al., 2012*). Intriguingly, ASCL1 levels also vary between human glioblastoma samples and low levels are correlated with shorter survival time (*Park et al., 2017*). Increasing ASCL1 levels was shown to promote terminal differentiation and attenuate tumorigenicity. Based on our results, we would predict that glioblastoma cells with high levels of TLX would exhibit low ASCL1 expression.

To determine if TLX and ASCL1 expression are mutually exclusive in glioblastoma, we analysed a previously published single-cell RNA sequencing (scRNA seq) data set that profiled glioblastoma samples from 28 patients, including both adult and pediatric tumours (*Neftel et al., 2019*). Our analysis identified 7,835 single cells in this data set and, following cluster annotation based on previously known markers (*Neftel et al., 2019*), we found that expression of both TLX and ASCL1 was restricted to the malignant glioblastoma cells (6,766 cells) (*Figure 7A–A'*). However, when we compared TLX and ASCL1 expression within the malignant population, we found very few cells that expressed both transcripts (*Figure 7B*), suggesting that TLX and ASCL1 are indeed mutually exclusive in malignant glioblastoma cells.

## Discussion

Our results revealed the mechanism through which high levels of the orphan nuclear receptor Tll initiate tumours in the *Drosophila* CNS (*Figure 8A*). We showed that Tll is expressed in Type II NSCs during larval development, where it is required for Type II NSC identity and subsequent lineage progression. In the absence of Tll, the proneural transcription factor Ase is derepressed in Type II NSCs. As a consequence, transit amplifying INPs are no longer generated and the resulting NSC lineages have a lower neurogenic potential.

A recent study examined the role of Tll in embryonic brain and showed that *tll* mutant embryos lack Type II NSCs (*Curt et al., 2019*). However, it was shown many years ago that *tll* mutant embryos fail to generate many NCSs, not just Type II NSCs, due to lack of *l'sc* expression that precedes NSC delamination (*Younossi-Hartenstein et al., 1997*). As a result, *tll* null mutants are not viable and the effect of *tll* loss of function on Type II NSCs specifically has not been addressed.

In mice, TLX is expressed in NSCs during embryonic development and in adulthood (*Li et al., 2012*; *Li et al., 2008*; *Liu et al., 2008*; *Shi et al., 2004*). Embryonic NSCs display defects in proliferation in the absence of TLX (*Li et al., 2008*) and the loss of TLX in adults results in the loss of transit-amplifying intermediates and reduction in neurogenesis (*Li et al., 2012*; *Liu et al., 2008*; *Niu et al., 2011*; *Shi et al., 2004*). While these effects were previously attributed to changes in the NSC cell cycle, our results suggest a cell fate change may occur due to the loss of TLX.

High levels of TLX in human glioblastoma are correlated with tumour aggressiveness (*Park et al., 2010*; *Zou et al., 2012*). High level expression of TLX results in glioblastoma-like lesions derived from SVZ NSC lineages in mouse models of glioblastoma (*Liu et al., 2010*) indicating that TLX can also promote glioblastoma development. However, it was not known how high TLX leads to glioblastoma, nor had the cellular origin of TLX-induced tumours been identified. TLX and its *Drosophila* homologue, Tll, are highly conserved proteins (*Yu et al., 1994*) and we found that both genes are able to revert INPs to NSC fate as a first step in tumour initiation. Ectopic expression of Tll was also sufficient to induce the expansion of NSCs throughout the *Drosophila* CNS, demonstrating the widespread vulnerability of NSC and progenitor populations to ectopic Tll expression.

We found that the ectopic NSCs resulting from high Tll expression are negative for Ase. We showed that Tll binds to the *ase* locus, suggesting that Tll directly represses *ase*. The absence of Ase

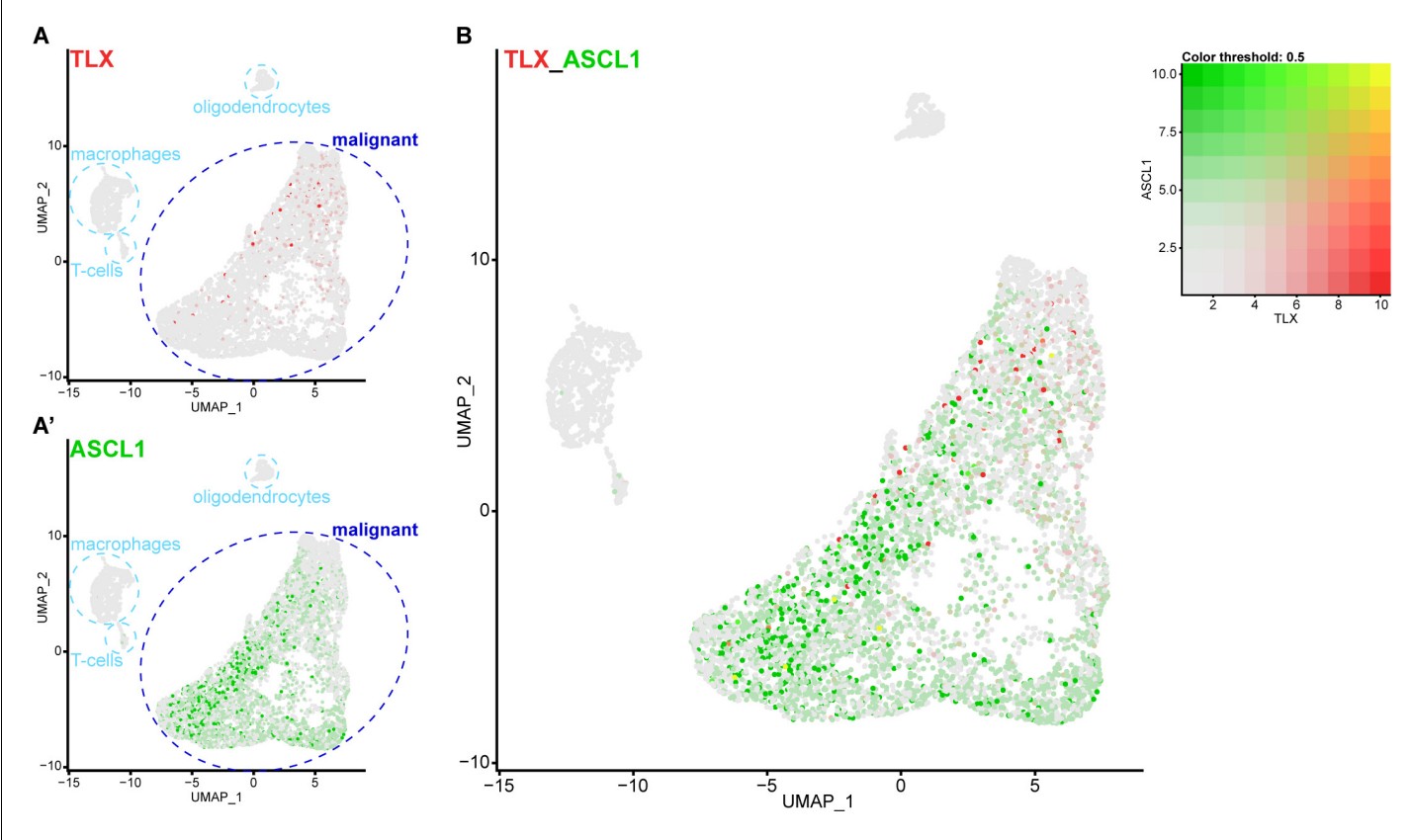

**Figure 7.** Single cell RNA sequencing reveals that TLX and ASCL1 appear to be mutually exclusive in human glioblastoma. (**A**) Uniform Manifold Approximation and Projection (UMAP) plot of 7,835 single cells coloured by TLX expression (red). Clusters were annotated based on previously known markers. TLX expression is only detected in the malignant cells. (**A'**) UMAP plot coloured by ASCL1 expression (green). ASCL1 expression is only detected in the malignant cells. (**B**) UMAP plot coloured by expression of both TLX (red) and ASCL1 (green), which appear mutually exclusive. Yellow indicates cells that express high levels of both TLX and ASCL1. Single cell RNA sequencing data and cluster markers obtained from *Neftel et al. (2019)*.

is a hallmark of Type II NSCs. Therefore, ectopic Tll promotes a cell fate change from INP/Type I NSC to Type II NSC and thereby initiates tumourigenesis.

The capacity of Tll to induce NSC expansion had been reported previously as part of a study showing that Tll regulates the proliferation of larval mushroom body NSCs and GMCs (*Kurusu et al., 2009*). The authors showed that overexpressing Tll resulted in ectopic NSCs, but they did not identify the origin of these tumours and argued against a role for Tll in Type II NSC fate (*Kurusu et al., 2009*). Tll-induced tumourigenesis could be blocked by ectopic expression of Pros (*Kurusu et al., 2009*). However, ectopic Pros results in the loss of NSCs even in wild type brains (*Cabernard and Doe, 2009*). In contrast, Type I NSC lineages appear normal after Ase misexpression in wild type brains (*Bowman et al., 2008*). Furthermore, it has been reported that high levels of the human homologue of Pros, PROX1, exacerbate glioblastoma (*Elsir et al., 2010*; *Goudarzi et al., 2018*; *Roodakker et al., 2016*; *Xu et al., 2017*), arguing against PROX1 expression as a therapeutic strategy.

We found that the tumourigenic capacity of *Drosophila* Tll and human TLX was highly conserved (*Figure 8B*). Human TLX could also induce ectopic Type II NSCs from INPs through the repression of Ase. Analysis of scRNA seq from glioblastoma revealed that TLX and ASCL1 expression is mutually exclusive. It is notable that the origin of human glioblastoma has been mapped to the SVZ (*Lee et al., 2018*). While TLX positive NSCs have been identified in both the SVZ and dentate gyrus, high levels of TLX giving rise to glioblastoma has only been shown robustly in the SVZ (*Liu et al., 2010*). Furthermore, a recent study demonstrated that low expression levels of ASCL1 correlate with glioblastoma malignancy (*Park et al., 2017*). Ectopic expression of ASCL1 in glioblastoma stem cells

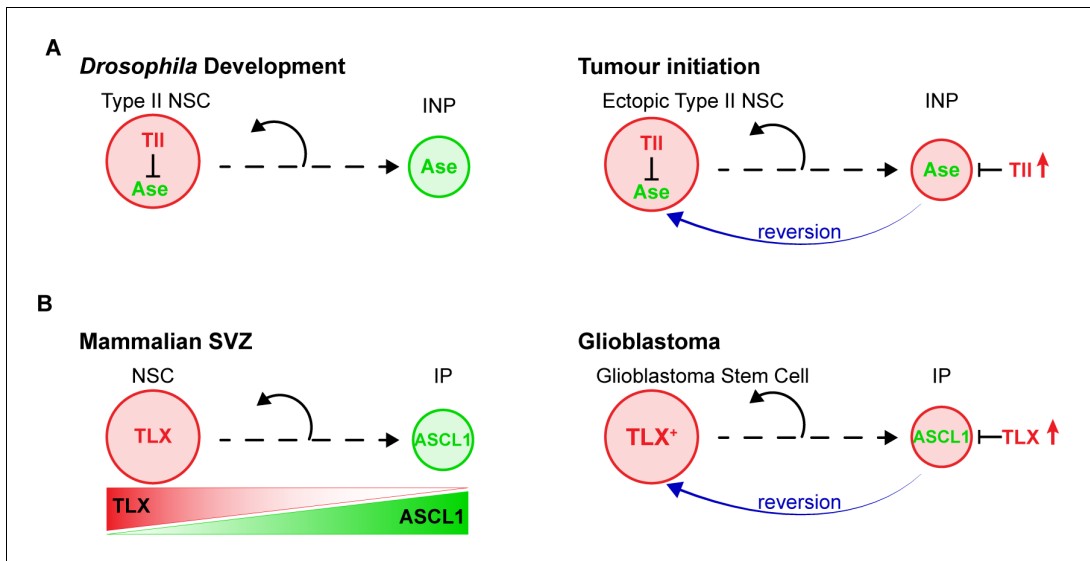

**Figure 8.** Model – Tll reverts INPs to NSC fate to initiate tumourigenesis. (**A**) Schematics depicting the promotion of Type II NSC fate by Tll (red) in development and tumourigenesis. Tll must be down regulated in Type II lineages to allow differentiation. Ase (green) expression is activated during differentiation. If Tll is high in INPs, or in Type I lineages, Type II NSC fate is maintained, or induced, and tumours form. (**B**) In the adult mouse SVZ, TLX expression is high in NSCs and lower in intermediate progenitors (IPs) (*Li et al., 2012*; *Obernier et al., 2011*), whereas ASCL1 is high in IPs and low in NSCs (*Kim et al., 2011*; *Parras et al., 2004*). Based on our results, we predict that high levels of TLX associated with aggressive glioblastoma revert IPs through the repression of ASCL1 to promote the generation of glioblastoma stem cells.

was sufficient to promote neuronal differentiation. Based on our results in *Drosophila*, we predict that introducing ASCL1 would override the repressive effect of TLX, induce neuronal differentiation and reduce tumour growth, thereby providing an effective treatment.

Our results indicate that INPs are the tumour initiating cells in Type II NSC lineages expressing high levels of the orphan nuclear receptor Tll and potentially implicate intermediate progenitors as one of the cells of origin in TLX+ glioblastomas, an aggressive and lethal brain tumour. We found that Ase is a direct target of Tll and that Ase expression not only blocks Tll-induced tumourigenesis, but also reinstates a normal neural differentiation programme.

## Materials and methods

### Key resources table

| Reagent type (species) or resource | Designation | Source or reference | Identifiers | Additional information |
|---|---|---|---|---|
| Genetic reagent (*D. melanogaster*) | w[1118];+;+ | BDSC | RRID:BSDC_3605 | |
| Genetic reagent (*D. melanogaster*) | Ay-GAL4, UAS-*GFP* | BDSC | RRID:BDSC_4411 | |
| Genetic reagent (*D. melanogaster*) | Ay-GAL4, UAS-*lacZ(nls)* | BDSC | RRID:BDSC_4410 | |
| Genetic reagent (*D. melanogaster*) | btd-GAL4 | (*Estella et al., 2003*) | | |
| Genetic reagent (*D. melanogaster*) | erm-GAL4 | BDSC | RRID:BDSC_40731 | GMR9D11-GAL4 |
| Genetic reagent (*D. melanogaster*) | GMR71C09-GAL4 | BDSC | RRID:BDSC_39575 | |

*Continued on next page*

*Continued*

| Reagent type (species) or resource | Designation | Source or reference | Identifiers | Additional information |
|---|---|---|---|---|
| Genetic reagent (*D. melanogaster*) | insc-GAL4 | (*Luo et al., 1994*) | | GAL4[MZ1407] |
| Genetic reagent (*D. melanogaster*) | pntP1[14-94]-GAL4 | (*Zhu et al., 2011*) | | |
| Genetic reagent (*D. melanogaster*) | wor-GAL4 | (*Albertson et al., 2004*) | | |
| Genetic reagent (*D. melanogaster*) | OK371-GAL4 | BDSC | RRID:BDSC_26160 | VGlut[OK371] |
| Genetic reagent (*D. melanogaster*) | tub-GAL80[ts] | BDSC | RRID:BDSC_7018 | |
| Genetic reagent (*D. melanogaster*) | UAS-*ase* | (*Brand et al., 1993*) | | |
| Genetic reagent (*D. melanogaster*) | UAS-*FLP* | BDSC | RRID:BDSC_4539 | |
| Genetic reagent (*D. melanogaster*) | UAS-*FLP* | BDSC | RRID:BDSC_4540 | |
| Genetic reagent (*D. melanogaster*) | UAS-*lacZ* | (*Brand and Perrimon, 1993*) | | |
| Genetic reagent (*D. melanogaster*) | UAS-*LT3-NDam* | (*Southall et al., 2013*) | | |
| Genetic reagent (*D. melanogaster*) | UAS-*LT3-NDam-tll* | this study | | Tll-Dam fusion for Targeted DamID |
| Genetic reagent (*D. melanogaster*) | UAS-*mCD8-GFP* | BDSC | RRID:BDSC_5130 | |
| Genetic reagent (*D. melanogaster*) | UAS-*mCD8-GFP* | BDSC | RRID:BDSC_5137 | |
| Genetic reagent (*D. melanogaster*) | UAS-*mCD8-mCherry* | BDSC | RRID:BDSC_27391 | |
| Genetic reagent (*D. melanogaster*) | UAS-*myr-mRFP* | BDSC | RRID:BDSC_7118 | |
| Genetic reagent (*D. melanogaster*) | UAS-*myr-mRFP* | BDSC | RRID:BDSC_7119 | |
| Genetic reagent (*D. melanogaster*) | UAS-*tll* | Kyoto DGRC | 109680 | |
| Genetic reagent (*D. melanogaster*) | UAS-*tll*-miRNA[s] | (*Lin et al., 2009*) | | |
| Genetic reagent (*D. melanogaster*) | UAS-tll-shRNA | VDRC | 330031 | |
| Genetic reagent (*D. melanogaster*) | UAS-*TLX* | this study | | Human TLX under the control of UAS |
| Genetic reagent (*D. melanogaster*) | G-TRACE | BDSC | RRID:BDSC_28280 | |
| Genetic reagent (*D. melanogaster*) | G-TRACE | BDSC | RRID:BDSC_28281 | |
| Genetic reagent (*D. melanogaster*) | erm-CD4-tdTomato | (*Han et al., 2011*) | | R9D11-CD4-tdTomato |
| Genetic reagent (*D. melanogaster*) | erm-mCD8-GFP | (*Zhu et al., 2011*) | | R9D11-mCD8-GFP |
| Genetic reagent (*D. melanogaster*) | erm-lacZ | (*Haenfler et al., 2012*) | | R9D11-*lacZ* |
| Genetic reagent (*D. melanogaster*) | Tll-GFP | BDSC | RRID:BDSC_30874 | |

*Continued on next page*

*Continued*

| Reagent type (species) or resource | Designation | Source or reference | Identifiers | Additional information |
|---|---|---|---|---|
| Genetic reagent (*D. melanogaster*) | Pnt-GFP | BDSC | RRID:BDSC_42680 | |
| Genetic reagent (*D. melanogaster*) | FRT82B, *tub*-GAL80 | BDSC | RRID:BDSC_5135 | |
| Genetic reagent (*D. melanogaster*) | FRT82B | BDSC | RRID:BDSC_2035 | |
| Genetic reagent (*D. melanogaster*) | FRT82B,*tll*^l49/TM6B | (*Pignoni et al., 1990*) | | |
| Genetic reagent (*D. melanogaster*) | dpn>KDRTs-stop-KDRTs>GAL4 | (*Yang et al., 2016*) | | |
| Genetic reagent (*D. melanogaster*) | ase-GAL80 | (*Neumüller et al., 2011*) | | |
| Genetic reagent (*D. melanogaster*) | stg14-*kd* | (*Yang et al., 2016*) | | |
| Antibody | rabbit anti-Ase (polyclonal) | (*Brand et al., 1993*) Gift from the Jan Lab | | IF 1:2,000 |
| Antibody | chicken anti-β-Galactosidase (polyclonal) | abcam | ab9361 | IF 1:1,000 |
| Antibody | rabbit anti-β-Galactosidase (polyclonal) | Cappel (now MP Biomedicals) | 55976 | IF 1:10,000 |
| Antibody | guinea pig anti-Dpn (polyclonal) | (*Caygill and Brand, 2017*) | | IF 1:5,000 |
| Antibody | rat anti-Elav (monoclonal) | DSHB | 7E8A10 conc. | IF 1:100 |
| Antibody | chicken anti-GFP (polyclonal) | abcam | ab13970 | IF 1:2,000 |
| Antibody | rabbit anti-PntP1 (polyclonal) | A gift from Jim Skeath | | IF 1:500 |
| Antibody | mouse anti-Pros (monoclonal) | DSHB | MR1A | IF 1:30 |
| Antibody | rabbit anti-Tll (polyclonal) | (*Kosman et al., 1998*) Asian Distribution Center for Segmentation Antibodies | | IF 1:300 |

## Fly stocks and husbandry

*Drosophila melanogaster* were reared in cages at 25 °C. Embryos were collected on yeasted apple juice plates. For experiments involving GAL80^ts embryos were kept at 18 °C until hatching. After hatching, larvae were transferred to a yeasted food plate and reared to the desired stage before dissection. Please see *Supplementary file 1* for experimental genotypes and the temperature at which larvae were raised for each experiment.

The following GAL4 lines were used: Ay-GAL4, UAS-*GFP* (BL4411), Ay-GAL4, UAS-*lacZ(nls)* (BL4410), btd-GAL4 (*Estella et al., 2003*), erm-GAL4 (*Pfeiffer et al., 2008*; *Weng et al., 2010*) (R9D11-GAL4, BL40731), GMR71C09-GAL4 (*Li et al., 2014*) (BL39575), insc-GAL4 (GAL4^MZ1407) (*Luo et al., 1994*), pntP1^14-94-GAL4 (*Zhu et al., 2011*), wor-GAL4 (*Albertson et al., 2004*), OK371-GAL4 (*VGlut*^OK371-GAL4) (BL26160). *tub*-GAL80^ts (BL7018) was used to restrict GAL4 activity to larval stages as indicated.

The following UAS-transgenes were used: UAS-*ase* (*Brand et al., 1993*), UAS-*FLP* (BL4539 and BL4540), UAS-*lacZ* (*Brand and Perrimon, 1993*), UAS-*LT3-NDam* (*Southall et al., 2013*) and UAS-*LT3-NDam-tll* (this study), UAS-*mCD8-GFP* (BL5130 and BL5137), UAS-*mCD8-mCherry* (BL27391), UAS-*myr-mRFP* (BL7118 and BL7119), UAS-*tll* (*Kurusu et al., 2009*) (Kyoto Stock Center 109680),

UAS-*tll*-miRNA[s] (*Lin et al., 2009*), UAS-*tll*-shRNA (VDRC 330031), UAS-*TLX* (this study), G-TRACE (BL28280 and BL28281). *w*[1118] was used as a reference stock.

The following reporter lines were used: erm-CD4-tdTomato (R9D11-CD4-tdTomato) (*Han et al., 2011*), erm-mCD8-GFP (R9D11-mCD8-GFP) (*Zhu et al., 2011*), erm-lacZ (R9D11-lacZ) (*Haenfler et al., 2012*) and Tll-GFP (*Venken et al., 2009*) (BL30874). Tll-GFP is a protein fusion under the control of a 20 kb insert containing the *tll* coding sequence and surrounding regulatory sequences. Importantly, this construct can rescue the lethality of homozygous *tll*[l49] mutants (*data not shown*). Pnt-GFP (BL42680) is a protein fusion under the control of a 90.7 kb insert containing the *pnt* coding sequence and surrounding regulatory sequences. This construct can rescue the lethality of *pnt* amorphic heteroallelic combinations (*Boisclair Lachance et al., 2014*).

For MARCM clone analysis, virgin female flies carrying hsFLP[122]; wor-GAL4, UAS-mCD8-mCherry/ (CyOact-GFP); FRT82B, tub-GAL80 were crossed to male flies carrying w; erm-lacZ; FRT82B or w; erm-lacZ; FRT82B, *tll*[l49]/TM6B. *tll*[l49] is strong *tll* point mutation that is homozygous embryonic lethal (*Pignoni et al., 1990*). Embryos were collected on apple juice plates at 25°C and newly hatched larvae were transferred to yeasted food plates and raised at 25°C. Clones were induced by a heat shock in a water bath (5 min 37°C, 5 min rest at room temperature, 1 hr 37°C) at 24 hr ALH and larvae were dissected 72 hr later.

## Immunostaining

Brains were dissected in PBS, fixed in 4% formaldehyde/PBS for 20 min at room temperature and washed with PBS with 0.3% TritonX-100 (PBTx). Samples were blocked with 10% normal goat serum before overnight incubation with the following antisera: rabbit anti-Ase 1:2,000 (*Brand et al., 1993*) (a gift from the Jan lab), chicken anti-β-Galactosidase 1:1,000 (abcam ab9361), rabbit anti-β-Galactosidase 1:10,000 (Cappel), guinea pig anti-Dpn 1:5,000 (*Caygill and Brand, 2017*), rat anti-Elav 1:100 (DSHB, 7E8A10 conc.), chicken anti-GFP 1:2,000 (abcam ab13970), rabbit anti-PntP1 (1:500) (a gift from the Jim Skeath), mouse anti-Pros 1:30 (DSHB, MR1A), rabbit anti-Tll 1:300 (*Kosman et al., 1998*). Secondary antibodies conjugated to Alexa-405, Alexa-488, Alexa-546, Alexa-568, Alexa-633 all 1:500 (Life Technologies) or DyLight-405 1:200 (Jackson Laboratories) were used. Samples were mounted in Vectashield (Vector Laboratories) for imaging.

## *tll* RNA FISH

A set of 38 Stellaris FISH probes was designed against the *tll* coding sequence and labeled with Quasar 570. Third instar larval brains were fixed in 4% formaldehyde/PBS for 45 min at room temperature and then permeabilized in 70% ethanol/PBS for 6 hr at 4 °C. Brains were washed with Wash Buffer (10% formamide, 2xSSC) for 5 min before being incubated with probes (125 nM) in hybridisation buffer (100 mg/mL dextran sulfate, 10% formamide, 2xSSC) overnight at 45 °C. Brains were washed with Wash Buffer, stained with DAPI and mounted in Vectashield (Vector Laboratories) for imaging.

## Image acquisition and processing

Fluorescent images were acquired using a Leica SP8 confocal microscope. Images were analysed using Fiji (*Schindelin et al., 2012*), which was also used to adjust brightness and contrast in images. Adobe Illustrator was used to compile figures.

## Quantification and statistical analysis

GraphPad Prism version 7.00 for Mac OS X (www.graphpad.com) was used for statistical analysis. No data were excluded.

## Sequence alignment of human TLX and *Drosophila* Tll

Sequence alignment performed using EMBOSS Needle (https://www.ebi.ac.uk/Tools/psa/emboss_needle/) and UniProt alignment (https://www.uniprot.org/align/) tools.

## Generation of UAS-*TLX*

The coding sequence of human *TLX* (*Jackson et al., 1998*) was amplified from cDNA prepared from H9 ESCs (a kind gift from T. Otani) using the primers fwd: 5'-AGATGAATTCA

TGAGCAAGCCAGCCGG-3' and rev: 5'-ATGACTCGAGTTAGATATCACTGGATTTGTACATATC TGAAAGCAGTC-3'. The amplified product was cloned (using restriction enzymes EcoR1 and XhoI) into pUAST-attB and then integrated into attP40 by standard methods.

## Generation of UAS-LT3-NDam-*tll*

The coding sequence of *tll* was amplified from an embryonic cDNA library using the primers fwd: 5'-cagaaactcatctctgaagaggatctgcgagatctaATGCAGTCGTCGGAGG-3' and rev: 5' acagaagtaaggttcctt-cacaaagatcctctagaTCAGATCTTGCGCTGACT 3'. The amplified product was cloned via Gibson assembly into pUASTattB-LT3-NDam (*Southall et al., 2013*) cut with BglII and XbaI and then integrated into attP40 by standard methods.

## Targeted DamID

We used a recombinase-dependent system to restrict GAL4 expression to Type II lineages (*Yang et al., 2016*). *dpn* >KDRTs-stop-KDRTs>GAL4; *ase*-GAL80/CyO*act*-GFP; + virgins were crossed to *w*; UAS-*LT3-NDam-tll*; *stg*14-*kd* or *w*; UAS-*LT3-NDam*; *stg*14-*kd* males at 25 ˚C. Larvae were transferred to yeasted food plates within an hour of hatching and dissected 50 hr later. Analysis was performed using the damidseq_pipeline as described previously (*Marshall et al., 2016*). *dpn* >KDRTs-stop-KDRTs>GAL4 and *stg*14-*kd* flies were provided by T. Lee (*Yang et al., 2016*) and *ase*-GAL80 flies by J. Knoblich (*Neumüller et al., 2011*). DamID analysis was performed as described previously (*Marshall and Brand, 2015*) and the Integrative Genomics Viewer (IGV, version 2.3.68) was used to visualise binding tracks aligned to release 6 of the *Drosophila* genome.

## RNA single cell sequencing analysis

Single cell sequencing analysis was performed using Seurat version 3. Data was obtained from *Neftel et al. (2019)*, which was made available through the Broad Institute Single-Cell Portal (https://portals.broadinstitute.org/ single_cell/study/SCP393/single-cell-rna-seq-of-adult-and-pediatric-glioblastoma) and the Gene Expression Omnibus (GEO: GSE131928).

# Acknowledgements

We should like to thank L Jan and Y N Jan, J Knoblich, M Kurusu, C-Y. Lee, T Lee, T Otani, J Skeath, S Zhu, the Asian Distribution Centre for Segmentation Antibodies, Bloomington *Drosophila* Stock Centre, Developmental Studies Hybridoma Bank (DSHB), the Kyoto Stock Center (DGRC), and the Vienna*Drosophila*Resource Center (VDRC) for reagents. We thank D J Kunz for advice on glioblastoma single cell RNA sequencing data, LYJ Tang for performing single cell RNA sequencing analysis and R Krautz for analyzing the Tll TaDa binding data. We thank L Otsuki for helpful discussions.

This work was funded by Wellcome Trust Senior Investigator Award (103792 to AHB) and the Royal Society Darwin Trust Research Professorship (to AHB) and Wellcome Trust PhD Studentship (102454 to AEH). AHB acknowledges core funding to The Gurdon Institute from the Wellcome Trust (092096) and CRUK (C6946/A14492).

# Additional information

## Funding

| Funder | Grant reference number | Author |
|---|---|---|
| Wellcome | 103792 | Andrea H Brand |
| Royal Society | | Andrea H Brand |
| Wellcome | 102454 | Anna E Hakes |
| Wellcome | 092096 | Andrea H Brand |
| Cancer Research UK | C6946/A14492 | Andrea H Brand |

The funders had no role in study design, data collection and interpretation, or the decision to submit the work for publication.

## Author contributions
Anna E Hakes, Conceptualization, Resources, Data curation, Formal analysis, Investigation, Methodology; Andrea H Brand, Conceptualization, Resources, Formal analysis, Supervision, Funding acquisition, Investigation, Project administration

## Author ORCIDs
Anna E Hakes (iD) https://orcid.org/0000-0002-8664-1014
Andrea H Brand (iD) https://orcid.org/0000-0002-2089-6954

## Decision letter and Author response
Decision letter https://doi.org/10.7554/eLife.53377.sa1
Author response https://doi.org/10.7554/eLife.53377.sa2

# Additional files

## Supplementary files
• Supplementary file 1. *Drosophila* genotypes and experimental conditions.
• Transparent reporting form

## Data availability
All data generated or analysed during this study are included in the manuscript and supporting files.

The following previously published dataset was used:

| Author(s) | Year | Dataset title | Dataset URL | Database and Identifier |
| --- | --- | --- | --- | --- |
| Neftel C, Laffy J, Filbin MG, Hara T | 2019 | single cell RNA-seq analysis of adult and paediatric IDH-wildtype Glioblastomas | https://www.ncbi.nlm.nih.gov/geo/query/acc.cgi?acc=GSE131928 | NCBI Gene Expression Omnibus, GSE131928 |

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
