## [Decision Letter]

**Acceptance summary:**

This paper demonstrates very elegantly that the transcription factor Tailless that is expressed in type II neuroblasts is required to repress Asense (the ortholog of ASCL1), and that its loss leads to the switch of identity from type II to type I neuroblasts. Interestingly, the paper shows that tumorgenesis resulting from overexpression of Tll is due to the reversion of intermediate neural progenitors to dividing neuroblasts, and that this is due to the Ase down-regulation. But the most exciting point of this paper is the demonstration that these brain tumors can be avoided by restoring Ase expression, which might be an important tool to eventually address why human glioblastoma have a poor prognosis when ASCL1 is also affected.

**Decision letter after peer review:**

Thank you for submitting your article "Tailless/TLX reverts intermediate neural progenitors to stem cells driving tumourigenesis via repression of *asense/ASCL1*" for consideration by *eLife*. Your article has been reviewed by three peer reviewers, and the evaluation has been overseen by a Reviewing Editor and Utpal Banerjee as the Senior Editor. The following individuals involved in review of your submission have agreed to reveal their identity: Yan Song (Reviewer #1).

The reviewers have discussed the reviews with one another and the Reviewing Editor has drafted this decision to help you prepare a revised submission.

In short, the three reviewers are very positive about the paper and its relevance to human cancer.

One of the reviewers mentioned that the fact that overexpression of Tll causes brain tumors was known and so was its role in Type II/Type I neuroblasts, in particular with a recent paper from the Thor lab in *eLife*. However, all three reviewers are conscious of the mechanistic insights that your paper provides, and, more importantly, the connection to human brain tumors and the different prognosis depending on the levels of ASCL1/Ase.

However, before proceeding with publication, the reviewers would like you to add two points, which you should be able to achieve in the coming two months:

– One is to present the DamID data in more detail, in particular genes other than Ase that might play a role in the process.

– The reviewers would also like you to explore the link between Tll and the Notch pathway since some of the phenotypes are quite similar. It should be fairly simple in your hands to look at interactions between these pathways.

Reviewer #1

The orphan nuclear receptor TLX is an important neural stem cell regulator in both normal and tumorigenic conditions. TLX is expressed in neural stem cells (NSCs) during mouse embryonic development and in adulthood and is required for neurogenesis. High levels of TLX have been detected in glioblastoma and correlate with poor patient prognosis. However, the cellular origin and the molecular mechanisms underlying high TLX-induced brain tumor formation remain unclear. In the study, the authors used *Drosophila* larval brain neural stem cell lineages as model system to tackle these important questions. The authors showed that fly TLX homolog, Tailless (Tll), is specifically expressed in type II NSCs. Downregulation of Tll led to upregulation of Asense (Ase; fly homolog of ASCL1) and a NSC identity switch from type II to type I. The authors further showed that high levels of Tll induced tumorigenesis by reverting intermediate neural progenitors (INPs) to a NSC state and identified INPs as the cellular origin of Tll-induced tumors. More importantly, the authors identified Ase as a key direct target of Tll. Ase is downregulated in high Tll-induced brain tumors and restoring Ase expression prevented Tll-induced tumorigenesis and reinstated normal neurogenesis. Finally, the authors suggested that such mutually exclusive relationship between Tll and Ase holds true in human glioblastoma samples via analysis of single-cell RNA-seq.

Overall, this is a very interesting study of potential therapeutic values. Most experiments are well-controlled and well performed. However, before recommending publication the following points need be addressed:

1) Through DamID-seq, the authors identified Ase as the major target of Tll in NSC lineages. Indeed, high Tll-induce tumor could be completely blocked by Ase co-expression. However, since Ase inactivation itself does not lead to tumor formation (Bowman et al., 2008), downregulation of Ase seems to be a permissive but not sufficient step in Tll-induced tumorigenesis. Therefore, other self-renewal or differentiation gene(s) is very likely to be important target gene of Tll in NSC lineages. Can the authors re-examine their DamID-seq results and find out such important target gene(s)?

2) Figure 4 showed that Tll can induced tumor from type I NSCs. However, downregulation of Ase alone might lead to a type I to type II NSC identity switch, but not tumorigenesis (Bowman et al., 2008). What is the cellular origin and molecular mechanisms underlying Tll-induced tumorigenesis in type I NSC lineages?

3) Whether Ase, in turn, inhibits Tll expression in INPs or type I NSCs?

Reviewer #2

They identify a role for *tll* in the developing *Drosophila* CNS, in Type II neuroblasts (NBs). They identify *ase* as a key target of *tll*, and find that simultaneous overexpression of *ase* can suppress the *tll* overexpression effects. They furthermore use DamID to identify *ase* as a putative direct target of Tll. These are very interesting findings. However, three major points temper my enthusiasm for this study.

Major points:

1) The role of *tll* in *Drosophila* NB biology is not novel. *tll* was previously shown to be important for brain NB generation (Younossi-Hartenstein et al., 1997) and MBNB proliferation (Kurusu et al., 2009). Even more importantly, *tll* was recently show to be expressed in Type II NBs in the embryo and necessary for their generation/development, and *tll* could act with *erm* to convert Type I NBs in the embryonic nerve cord to Type II NBs, as well as act in the developing wing disc to generate ectopic Type II NBs (Curt et al., 2019). Against this backdrop, the current study does take major strides in our understanding of Type II NB biology.

2) In the same vein, there is a large body of work on the NB->INP->GMC->Neuron transition in Type II NBs. This has identified key "gating" roles for Notch-Dpn-E(spl), as well as for Brm, Brat, Klu, PntP1, Btd, Erm, Mediator-Complex, Barc and Trx. However, the possible connection between *tll* and these other regulators is not addressed (at least 24 publications). In particular, the connection between Notch pathway and *tll* is certainly worthy of more scrutiny, as Notch signalling, similar to *tll*, acts in Type II to repress Ase and Erm expression, hence preventing NBs from prematurely becoming INPs. Moreover, there is growing connection between *tll*/Tlx and Notch: *tll* mutants show loss of expression of the proneural gene *l'sc* (Younossi-Hartenstein et al., 1997), which is negatively regulated by Notch; *tll* and Notch signalling intersect in the developing *Drosophila* embryonic optic placodes (Mishra et al., 2018, PLoS Genet.); the *C. elegans tll* orthologue *nhr-67* regulates both *lin-12* (Notch) and *lag-2* (Delta) during uterus development (Vergehese et al., 2011, Dev. Biol.); mouse *tll* orthologue Nr2E1 (aka Tlx) negatively regulates the canonical Notch target gene Hes1 (Luque-Molina et al., 2019, Stem Cell Reports). Hence, a key issue to address would be the intersection between *tll* and Notch, and given that NHRs are able to act both as transcriptional repressors and activators, two simple models emerge: (A) Tll acts with NotchICD-Su(H)-Mam on E(spl), and the obligate E(spl) repressors then act to directly repress *ase*, (B) alternatively, Tll acts combinatorially with E(spl) to directly repress *ase*. A straightforward way to test these two models would be to analyse expression of E(spl)-reporters in Tll LOF and GOF. In addition, it would be important to analyse Tll/*tll* expression in Notch pathway GOF and LOF.

3) DamID: Does Tll-Dam bind to other genes in the NB-INP-GMC-neuron pathway e.g., *pnt, erm, E(spl), btd, klu, pros, dpn*? Importantly, the evidence for Tll binding to the *ase* gene, but possibly also to *dpn* and *E(spl),* has bearing on the model for *tll* action. In addition, the DamID results are not described in great detail.

Reviewer #3

The Hakes and Brand manuscript contains detailed insightful information on the function of *Drosophila* Tailless. It covers both the role of Tll in normal neurogenesis and the tumourigenic effect of Tll overexpression.

The authors show that *tll* is required to maintain the Type II state: strikingly, upon *tll* loss, Type II neural stem cells are transformed into Type I, hence significantly compromising neurogenesis due to the lack of intermediate neural progenitors. They also show *tll* levels are critical: TLL over expression in the larval brain causes tumours derived from both Type II and Type I lineages. In the former, tumours develop as a consequence of intermediate progenitors reverting to a neural stem cell-like state. In the later, ectopic Tll results in the loss of Ase and in some cases even in the generation of intermediate neural progenitors, both suggestive of a certain degree of transdetermination from Type I towards Type II neural stem cells.

The manuscript goes on to demonstrate that Asense is a direct target of Tll repression during normal development as well as in the tumours that develop upon Tll upregulation to the extent that forcing Ase expression suppresses Tll-induced tumours

Overexpression of the human homologue TLX, which shares a remarkable level of protein sequence identity, cofactors, and function with *Drosophila* Tll, has been linked to the development of malignant astrocytomas. Interestingly, the authors find that TLX and ASCL1 (human Ase) are mutually exclusive in cells from human glioblastomas, which indeed is closely reminiscent of the results reported in the manuscript regarding fly neurogenesis and brain tumour development.

The reported studies take advantage of state-of-the-art techniques to perform sophisticated cell type-specific experimental manipulations to create the mutant conditions, and to trace the cell lineages of interest. The results are very well documented, presented, and discussed. Altogether, this is a high quality manuscript that I am happy to recommend for publication.

---

## [Author Response]

Reviewer #1[…]Overall, this is a very interesting study of potential therapeutic values. Most experiments are well-controlled and well performed. However, before recommending publication the following points need be addressed:1) Through DamID-seq, the authors identified Ase as the major target of Tll in NSC lineages. Indeed, high Tll-induce tumor could be completely blocked by Ase coexpression. However, since Ase inactivation itself does not lead to tumor formation (Bowman et al., 2008), downregulation of Ase seems to be a permissive but not sufficient step in Tll-induced tumorigenesis. Therefore, other self-renewal or differentiation gene(s) is very likely to be important target gene of Tll in NSC lineages. Can the authors reexamine their DamID-seq results and find out such important target gene(s)?

We have included the full list of Tll target genes as an excel file (Figure 6—figure supplement 1—source data 1). In addition to *ase,* Tll has multiple targets that could contribute to the Tll phenotypes we observe (including *pnt, dpn, klu, wor, cycA, cycB, cycE*).

2) Figure 4 showed that Tll can induced tumor from type I NSCs. However, downregulation of Ase alone might lead to a type I to type II NSC identity switch, but not tumorigenesis (Bowman et al., 2008). What is the cellular origin and molecular mechanisms underlying Tll-induced tumorigenesis in type I NSC lineages?

We showed that Tll must be downregulated in Type II lineages to allow differentiation to proceed. When expressed in Type I NSCs, Tll induces a cell fate change from Type I to Type II NSC and Tll is maintained at high levels in these transformed lineages. Once a Type I NSC has been transformed into a Type II NSC, differentiation is blocked due to continued high levels of Tll, which results in tumourigenesis. Tumourigenesis likely occurs through the inactivation of Ase in addition to other Tll target genes (see above).

3) Whether Ase, in turn, inhibits Tll expression in INPs or type I NSCs?

We show that ectopic expression of Ase in Type II NSCs does not repress Tll nor does expressing Ase in Tll-induced tumours. We have added these data as Figure 6—figure supplement 2. Therefore, Ase does not act directly on Tll but through other downstream targets.

Reviewer #2They identify a role for tll in the developing *Drosophila* CNS, in Type II neuroblasts (NBs). They identify ase as a key target of tll, and find that simultaneous overexpression of ase can suppress the tll overexpression effects. They furthermore use DamID to identify ase as a putative direct target of Tll. These are very interesting findings. However, three major points temper my enthusiasm for this study.Major points:1) The role of tll in *Drosophila* NB biology is not novel. tll was previously shown to be important for brain NB generation (Younossi-Hartenstein et al., 1997) and MBNB proliferation (Kurusu et al., 2009). Even more importantly, tll was recently show to be expressed in Type II NBs in the embryo and necessary for their generation/development, and tll could act with erm to convert Type I NBs in the embryonic nerve cord to Type II NBs, as well as act in the developing wing disc to generate ectopic Type II NBs (Curt et al., 2019). Against this backdrop, the current study does take major strides in our understanding of Type II NB biology.

*tll* null mutants are not viable and the effect of *tll* loss of function on Type II NSCs specifically has never been addressed. We have identified a previously unknown role for Tll in larval Type II NSCs. Our data show that Tll is necessary and sufficient to promote Type II NSC fate. In the absence of Tll, Type II NSCs switch to a Type I identity, whereas ectopic expression of Tll alone is sufficient to induce Type II NSC fate in Type I NSCs.

It was shown some years ago (Younossi-Hartenstein et al., 1997) that *tll* mutant embryos fail to generate many different types of NSC (not just Type II NSCs) due to lack of *l’sc* expression, which precedes NSC delamination. This would explain why Curt et al., 2019 found that *tll* mutant embryos lack Type II NSCs. Curt et al. promoted Type II NSC identity in the embryonic brain by co-expression of Tll and Erm, not Tll alone. Later in development, Tll is indeed required for the proliferation of larval mushroom body NSCs and GMCs (Kurusu et al., 2009), demonstrating that Tll has diverse roles in brain development.

We have added these points to our Discussion.

2) In the same vein, there is a large body of work on the NB->INP->GMC->Neuron transition in Type II NBs. This has identified key "gating" roles for Notch-Dpn-E(spl), as well as for Brm, Brat, Klu, PntP1, Btd, Erm, Mediator-Complex, Barc and Trx. However, the possible connection between tll and these other regulators is not addressed (at least 24 publications). […] Hence, a key issue to address would be the intersection between tll and Notch, and given that NHRs are able to act both as transcriptional repressors and activators, two simple models emerge: (A) Tll acts with NotchICD-Su(H)-Mam on E(spl), and the obligate E(spl) repressors then act to directly repress ase,

We show that Tll binds directly to the *ase* locus. Therefore, the simplest explanation is that Tll acts directly on *ase*. Furthermore, we found that expression of the Notch signalling reporter E(spl)-mγ-GFP (Almeida and Bray., 2005) remains unchanged in Tll LOF Type II NSCs (see Author response image 1). These data argue against the reviewer’s first model that Tll acts with NotchICD-Su(H)-Mam on E(spl). More importantly to the reviewer’s point is that the published data indicate that Notch signalling represses Ase indirectly in Type II NSCs and instead maintains Type II NSC fate through the repression of Erm (Li et al., 2016).

**Author response image 1. respfig1:** N signalling is maintained in Tll LOF Type II NSCs. E(spl)-mγ-GFP, a N signalling reporter (Almeida and Bray., 2005), is expressed in Control Type II NSCs (left panel, solid white outlines) and in Tll LOF Type II NSCs (*tll*-miRNA[s], right panel, solid white outlines). Dotted outlines highlight three Type II lineages identified by *pntP1*>*act*-GAL4 driving UAS-*lacZ(nls). n* = 13 brains for Control; *n* = 12 brains for *tll*-miRNA[s]. Brains dissected at the end of the second instar larval stage. The control panel is a projection of two 1 μm slices; the *tll*-miRNA[s] panel is a single section confocal images. Scale bars represent 15 µm.

(B) alternatively, Tll acts combinatorially with E(spl) to directly repress ase. A straightforward way to test these two models would be to analyse expression of E(spl)-reporters in Tll LOF and GOF.

We have shown that Tll GOF results in ectopic Type II NSCs. Since it is known that Type II NSCs normally have active N signalling (Zacharioudaki et al., 2012), we would expect to see an increase in E(spl)-mγ-GFP^+^ cells in Tll GOF lineages. Indeed, we found that ectopic Type II NSCs in Tll GOF lineages express the Notch signalling reporter E(spl)-mγ-GFP (Almeida and Bray., 2005) (see Author response image 2).

**Author response image 2. respfig2:** N signalling is active in ectopic NSCs in Tll GOF Type II lineages. E(spl)-mγ-GFP (green), a N signalling reporter (Almeida and Bray., 2005), is expressed in Control Type II NSCs (top panels). Type II NSCs (Dpn^+^ (red) and Ase^-^ (blue)) are indicated with white arrowheads and lineages are outlined in dotted white lines. Likewise, N signalling is active in the ectopic Type II NSCs in Tll GOF lineages (bottom panels). Note that N signalling is also active in Type I NSCs (Dpn^+^ and Ase^+^, see arrow in Tll OE zoom panels). Zoom panels are magnifications of the boxed regions shown. *btd-*GAL4,FRT19A/FM7*act*-GFP;+;*tub*-GAL80^ts^ virgins were crossed to *w; E(spl)-mγ-GFP*; UAS-*myr-mRFP*/TM6B (Control) or *w; E(spl)-mγ-GFP*; UAS-*tll*, UAS-*myr-mRFP*/TM6B (Tll OE) males. Dotted white lines indicate *btd*-GAL4>*myr-mRFP*. GAL4 activity was restricted to larval development and brains were dissected after 3 days at 29 °C. *n* = 4 for Control and *n* = 5 for Tll OE. Single section confocal images. Scale bars represent 15 µm.

In addition, it would be important to analyse Tll/tll expression in Notch pathway GOF and LOF.

It is known that N GOF in Type II lineages results in ectopic Type II NSCs (Bowman et al., 2008; Weng et al., 2010; Xiao et al., 2012; Song et al., 2011; Farnsworth et al., 2015) and we have shown that normal Type II NSCs express Tll. Therefore, an increase in Tll^+^ cells in N GOF Type II lineages would be consistent with previous literature and the data we have presented in this study.

There are some important differences between the Tll and N LOF phenotypes. We have shown that Type II NSCs become Type I NSCs without Tll, whereas Type II NSCs appear to undergo premature differentiation without N signalling (Li et al., 2016). Although it was reported previously that Type II NSCs switch to Type I NSCs without N signalling, Erm is precociously expressed in the NSC and maintained in the lineage, *i.e.* they are not Type I NSCs (Li et al., 2016). In contrast, *erm* expression is absent entirely from Tll LOF Type II lineages. Indeed, all Type II NSC and lineage features are absent in Tll LOF Type II NSCs (repression of Ase in NSC, plus PntP1 and *erm* expression). We have added the effect of Tll LOF on PntP1 expression as Figure 2—figure supplement 2C-D’.

We propose that Tll and N signalling act in parallel in Type II NSCs: Tll acts upstream to establish Type II NSC identity (including repression of *ase*) and that N signalling acts within Type II lineages to ensure timely differentiation of INPs (by repressing *erm* in Type II NSCs). However, as the reviewer states, there are many genes that regulate cell fate changes in Type II lineages. How Tll interacts with each of these genes will be an interesting topic for future study.

3) DamID: Does Tll-Dam bind to other genes in the NB-INP-GMC-neuron pathway e.g., pnt, erm, E(spl), btd, klu, pros, dpn? Importantly, the evidence for Tll binding to the ase gene, but possibly also to dpn and E(spl), has bearing on the model for tll action. In addition, the DamID results are not described in great detail.

We have included the full list of Tll target genes as an excel file (Figure 6—figure supplement 1—source data 1). In addition to *ase,* Tll as a transcription factor has multiple targets, including *pnt, erm, E(spl), btd, klu, pros* and *dpn*, that could contribute to the Tll phenotypes we observe.